# Running on Empty: A Metabolomics Approach to Investigating Changing Energy Metabolism during Fasted Exercise and Rest

**DOI:** 10.3390/metabo10100399

**Published:** 2020-10-08

**Authors:** Gavin Blackburn, Joshua Hay, Christine Skagen, Elizabeth Paul, Fiona Achcar, John Wilson, Cameron Best, Erin Manson, Karl Burgess, Michael P. Barrett, Jason M. R. Gill

**Affiliations:** 1Glasgow Polyomics, Wolfson Wohl Cancer Research Centre, College of Medical, Veterinary and Life Sciences, University of Glasgow, Glasgow G61 1BD, UK; Erin.Manson@glasgow.ac.uk (E.M.); Michael.Barrett@glasgow.ac.uk (M.P.B.); 2Center for Preventive Cardiology, Knight Cardiovascular Institute, Oregon Health & Science University, 3181 SW Sam Jackson Park Rd. Portland, OR 97239, USA; hayjo@ohsu.edu; 3Muscle Research Group, Department of Pharmacy, University of Oslo, Sem Sælands vei 3, 0371 Oslo, Norway; christine.skagen@farmasi.uio.no; 4BHF Glasgow Cardiovascular Research Centre (GCRC), Institute of Cardiovascular & Medical Sciences, University of Glasgow, 126 University Place, Glasgow G12 8TA, UK; lizziepaul@outlook.com (E.P.); John.Wilson@glasgow.ac.uk (J.W.); Jason.Gill@glasgow.ac.uk (J.M.R.G.); 5Wellcome Centre for Integrative Parasitology, Institute of Infection, Immunity and Inflammation, University of Glasgow, 126 University Place, Glasgow G12 8TA, UK; fiona.achcar@glasgow.ac.uk; 6Institute of Infection, Immunity and Inflammation, University of Glasgow, Sir Graeme Davis Building, 126 University Place, Glasgow G12 8TA, UK; c.best.1@research.gla.ac.uk; 7Institute of Quantitative Biology, Biochemistry and Biotechnology, The University of Edinburgh, Edinburgh EH9 3FF, UK; Karl.Burgess@ed.ac.uk

**Keywords:** energy metabolism, sports metabolism, nutritional metabolomics, biomarker research, liquid chromatography-mass spectrometry (LC-MS), fasted exercise

## Abstract

Understanding the metabolic processes in energy metabolism, particularly during fasted exercise, is a growing area of research. Previous work has focused on measuring metabolites pre and post exercise. This can provide information about the final state of energy metabolism in the participants, but it does not show how these processes vary during the exercise and any subsequent post-exercise period. To address this, the work described here took fasted participants and subjected them to an exercise and rest protocol under laboratory settings, which allowed for breath and blood sampling both pre, during and post exercise. Analysis of the data produced from both the physiological measurements and the untargeted metabolomics measurements showed clear switching between glycolytic and ketolytic metabolism, with the liquid chromatography-mass spectrometry (LC-MS) data showing the separate stages of ketolytic metabolism, notably the transport, release and breakdown of long chain fatty acids. Several signals, putatively identified as short peptides, were observed to change in a pattern similar to that of the ketolytic metabolites. This work highlights the power of untargeted metabolomic methods as an investigative tool for exercise science, both to follow known processes in a more complete way and discover possible novel biomarkers.

## 1. Introduction

The current understanding of metabolic processes influenced by exercise is based mainly on targeted analysis methods as described by Egan et al. [1]. Changes in over 200 metabolites were examined using liquid chromatography mass spectrometry (LC-MS) by Lewis et al. [2]. Significant changes were identified in several metabolites in 25 subjects whose blood was collected before and after the Boston Marathon. These included large increases in lipolysis and 3-hydroxybutyrate, the endpoint of ketolytic metabolism. Van Hall et al. have shown that during high intensity exercise glycolytic metabolism is raised, resulting in increases in lactate [3]. Targeted methods have been used to investigate other indicators of energy metabolism, such as the work by Gibala et al. showing increased flux in the tri-carboxylic acid (TCA) cycle in skeletal muscle during moderate exercise and after intense exercise [4]. While these methods have allowed for the analysis of markers of exercise and health, a large number of intermediate metabolites that are key to fully understanding the biochemistry present are not seen.

Using metabolomics, fundamental biochemical pathways involved in exercise can be probed using a single analytical platform, whereas targeted methods may require a wide range of analytical tools. The current state of metabolomics in sport and exercise science has been described by Heaney et al. [5], highlighting the use of targeted methods and exciting prospects of using untargeted metabolomics to identify biomarkers of health and performance effects of exercise intervention.

Untargeted metabolomics studies have shown elevated levels of lipid metabolism during training in endurance athletes and changes in acyl-carnitines have been linked to fatty acid oxidation [6,7,8]. While most studies currently focus on metabolites with a known function, untargeted metabolomics has also been used to identify metabolites with currently no known function related to exercise. For example, Malkar et al. identified changes in δ-valerolactam (2-piperidone) in response to exercise [9].

Interest in fasted exercising and ketolytic metabolism has grown in both research and the general public. While a large body of research exists comparing fasted exercise against fed exercise, there is currently a lack of research looking at the changes in the metabolome over time during fasted exercise and subsequent rest. Previous work investigated blood glucose levels in diabetic subjects during high-intensity interval training (HIIT) and moderate-intensity continuous training (MICT) after fasting [10]. From this work, it was revealed that there was no difference shown between pre and post HIIT and MICT for the length of time spent in the states of hyperglycaemia, hypoglycaemia or in a target glucose range over 24 h/during the nocturnal period. While this information is useful, it gives no insight into the other known energy processes which may in turn help to investigate cardiovascular health. Monitoring glucose levels for diabetics pre and post exercise provides valuable insights into how exercise can be effectively utilised to minimise the impact of the disease. However, this may not translate well to the general population where an individual’s glucose level is relatively stable.

This work seeks to utilise untargeted metabolomics methods to provide a much more comprehensive picture of the metabolism of fasted individuals both during and post-exercise. Metabolites associated with the key energy pathways, along with transporter metabolites, can be seen allowing for identification of switching between energy sources. The untargeted nature of the method also means that novel compounds, that are either contained in databases but not yet associated to energy metabolism, or are currently not found in databases, can be investigated.

## 2. Results

### 2.1. Physiological Testing

To assess the effort and recovery of the participants, heart rate (HR) was recorded at 15-min intervals and breath samples were collected every 15 min to calculate the percentage of VO_2_ max. As can be seen from Figure 1**,** average HR across the participants was similar in both the pre-exercise and rest phases. The same trend is observed for VO_2_ max measurements (Figure 2). These data indicate a similar exertion by all participants across the protocol and a similar level of endurance training. Baseline HR shows larger variation. The slight increase in variation of both HR and percentage of VO_2_ max measurements seen at 120 min is likely due to time trial pacing differences between competitive cyclists and athletes who use cycling for training.

Average respiratory exchange ratios (RERs) across all participants show an initial increase from the baseline in the exercise phase, then a downward trend with a sharp drop during rest. These values then remain below the baseline, showing a slight increase towards 240 min, Figure 3. Some variation in work rate was seen across participants but this was small and likely due to differing levels of glycogen in their muscles.

RER values of 0.7 or less are indicative of ketolytic metabolism. Values of 0.7 or less were seen for three participants and all participants were observed to approach this limit apart from participant 003.

### 2.2. Metabolomic Analysis

Metabolomic analysis identified 851 likely metabolite signals, with 632 annotated on the basis of mass and retention time prediction using IDEOM [11] and 51 metabolites identified against authentic standards [12] (see Appendix A). The implementation of a generalised linear model (GLM) identified 108 metabolites as significantly changing over time based on false discovery rate of less than 0.05 (S1). Due to the lack of replicates for individual subjects at each time point, the *p*-values for participant and the interaction term of time: participant were not presented. A heatmap was plotted for the data on the basis of fold-change comparisons to T0 to identify trends in metabolites. This can be seen in Figure 4.

### 2.3. Constant Load Phase

During the constant load phase, changes were observed across glycolytic metabolism. The profile of lactate over time resembles the physiological lactate concentrations measured in plasma and was seen to change significantly over the course of the protocol (corrected *p*-value = 1.6 × 10^−2^). Pyruvate, another metabolite of glycolysis, and malate, a key TCA intermediate (corrected *p*-value = 1.8 × 10^−2^ and 2.0 × 10^−3^ respectively). These metabolite profiles are consistent with glycolytic metabolism during the initial exercise phase.

At 60 min, several metabolite signals showed changes compared to the baseline based on fold change. These are Glycerol-3-phosphate (FC = 2.60) (identified based on authentic standard retention time), hexanoylcarnitine (3.41) and butanoylcarnitine (2.51), all metabolites involved in fatty acid transport and oxidation. These metabolites showed maximum fold changes against the baseline at 120 min and suggest a switch from glycolytic metabolism to ketolytic metabolism (Figure 5).

### 2.4. Time Trial Phase

Metabolomics samples were taken at the start and end of the time trial phase. Several key ketolytic metabolites were observed to increase against the baseline, including two key ketone body metabolic end points, acetoacetate (FC = 2.18, corrected *p*-value = 3.0 × 10^−16^) and 3-hydroxybutanoate (FC = 2.68, corrected *p*-value = 6.4 × 10^−15^). In addition to these, several fatty acids also showed increases, including hexadecanedioic acid (FC= 3.47, corrected *p*-value = 3.6 × 10^−6^), dodecanedioic acid (FC = 4.04, corrected *p*-value = 2.9 × 10^−5^) and 3-hydroxydodecanedioic acid (FC =4.35, corrected *p*-value = 1.9 × 10^−6^). Lactate, pyruvate and malate all returned to a FC of less than 2 by the end of the time trial phase (Figure 4). These changes are consistent with a switch from glycolytic metabolism to ketolytic metabolism. Previously detailed carnitines had maximum FCs at 120 min.

### 2.5. Recovery Phase

Further indication of ketolytic metabolism was observed during the recovery phase. Previously noted fatty acids showed large increases versus the baseline during recovery, with maximum differences observed at 150 min and decreasing at 180 and 240 min. Acetoacetate and 3-hydroxybutanoate continued to increase during the rest phase, both with maximum FCs at 240 min compared to the baseline (5.52 and 16.78, respectively). Hydroxybutyrylcarnitine followed the same pattern as 3-hydroxybutanoate (corrected *p*-value = 1.3 × 10^−15^). Previously noted carnitines continue to decrease across the rest phase to FCs of less than 2 by the end of the protocol. As carnitines are involved in fatty acid metabolism, by transporting them across the mitochondrial membrane, this decrease in carnitine levels follows a reduction in energy needs after the exercise phases have been completed.

### 2.6. Amino Acid Energy Metabolism

The alanine and glutamate levels showed small changes during the constant load phase of the protocol with alanine showing a maximum FC = 1.69 (corrected *p*-value = 1.6 × 10^−2^) at T30 and glutamate showing a maximum FC = 1.58 (corrected *p*-value = 2.8 × 10^−2^) at T60. The levels of both metabolites returned to approximately the baseline by T120 and remained at this level during the recovery phase. Arginine levels did not change through the course of the protocol. Glutamine, as the vehicle of nitrogen transport in the blood, was checked but was found not to change significantly across the experiment (corrected *p*-value = 0.97) This would indicate that amino acid energy metabolism is only occurring at a low level during glycolytic metabolism and during the recovery phase when ketolytic energy metabolism dominates [13,14] (Figure 4).

### 2.7. Novel Peptides Changing with Increasing Ketolytic Metabolites

Several signals in the data set were putatively annotated as small peptides (di-, tri and tetra-) that correlated with specific parts of the protocol. Of note, large increases in three possible peptides were seen during the rest phase, following a similar pattern to that of the ketolytic metabolites, with the largest changes in these signals deemed significant on the basis of the corrected *p*-value < 0.05. Initial increases in these signals were observed at T120. Two possible peptides increased across the first hour of the exercise protocol, with the largest change in one of these deemed significant (students *t*-test, uncorrected, compared to T0). Comparison of the predicted peptide sequence with known small gut peptides and larger peptides, such as cholecystokinin, were checked, but no matches were found.

## 3. Discussion

The physiological data shows that, while there is some variation, effort, in terms of HR and VO_2_ max, was consistent during the exercise phases of the protocol across the subjects. As this study uses a small number of participants, this physiological data is useful as it shows we can group participants together as biological replicates, a requirement of mass spectrometry metabolomic methods. As would be expected for trained endurance athletes, HR and VO_2_ max returned to the baseline by 135 min (after 15 min of rest). Average RER was seen at its lowest after 30 min of fasted rest, with several participants exhibiting an RER of less than 0.7, indicating ketolytic metabolism. There was a wide variation in RER between subjects at the end of the protocol, and it is therefore difficult to make an assessment of their metabolic state with respect to carbohydrate, fat and amino acid usage from the physiological data alone. As such, the use of metabolomics in this study has provided much greater insight into the participant’s fundamental energy metabolism.

Metabolomics has allowed a more detailed investigation into the energy metabolism of the subjects, as it allows us to track levels of metabolites in the blood over the course of the protocol. These metabolites highlight key changes in energy metabolism during the different phases of the experiment. On commencement of the constant load phase, changes in lactate and pyruvate, metabolites that are characteristic of glycolytic energy metabolism, increase rapidly, peaking at 15 min. The maximum lactate signal observed at 15 min indicates that this was the most intense period of exercise measured, where anaerobic energy metabolism is highest. Energy metabolism then becomes more aerobic during the last 30–45 min of the constant load phase and the time trial phase of the protocol. By 120 min, the levels of both lactate and pyruvate have reduced to less than 2-fold compared to the baseline and continue to drop across the rest phase. As these, and other metabolites detailed, are intracellular metabolites, care must be taken in the interpretation of this data. Metabolites such as pyruvate need to be assessed carefully, due to instability in plasma, whereas lactate is more stable. The changes observed are in line with what is known with regard to energy metabolism, with lactate increasing during the initial stages of the constant load phase, due to the operation of the Cori cycle.

Conversely, several metabolites that are indicative of the different stages of ketolytic metabolism are seen in the time trial phase and recovery phase of the protocol. Acyl-carnitines are seen to increase during the time trial phase. An initial increase in acyl-carnitines is required for ketolytic metabolism, as fatty acid oxidation occurs in the mitochondria so transport across the membrane must precede this. This is supported by the data as acyl-carnitine peaks appear, followed by fatty acids, with acetoacetate and 3-hydroxybutanoate increasing throughout the rest phase. A general increase in fatty acids and ketone bodies is seen during the time trial phase and continuing into the recovery phase with fatty acid levels, peaking around 150 min and the ketone body levels, specifically 3-hydroxybutanoate and acetoacetate, continuing to increase for the whole of the experimental protocol. Maximum signals for acyl-carnitines are seen at T120 and return to the baseline by T240. Observed fatty acid peaks reach a maximum at T150 and decrease from this point until T240. Acetoacetate and 3-hydroxybutanoate give maximum signals at T240. This cascade effect would seem to follow the dogmatic view that fatty acids must be transported to cells where they are broken down in the peroxisome. This shows that the switch to ketolytic metabolism occurs during exercise with substrates transported into the mitochondria and ketone body production beginning during the time trial phase and increasing at rest. These results are of interest to those hoping to reduce body fat, as glucose supplementation after exercise may cause a shift towards glycolytic metabolism [14,15].

Several possible small peptides were annotated in the data set. While care should be taken with these peaks as their shape/peak quality is poor and intensity is low, several showed correlations with well characterised energy metabolites from both the glycolytic and ketolytic energy pathways. Several small peptides are known to have important signalling roles, and larger peptides, such as cholecystokinin, which stimulates the digestion of fats and proteins, and ghrelin, which stimulates hunger and promotes fat storage when released in the gut, are important in appetite, hunger and metabolism, making these worth discussing. As different annotated peptides correlated with different parts of the exercise protocol, and different metabolites associated with different energy pathways, it is possible that they are involved in signalling which form of energy production to use. No peptidase inhibitors are added to the samples at any point during the metabolite extraction. It is therefore possible that the signals that change with the same pattern are fragments of larger peptides that undergo some breakdown prior to extraction. On the basis of the data presented here, it is not possible to determine the roles of these peptides.

The appearance of peptides during exercise and the following rest phase may also be indicative of muscle damage and breakdown. Previous work has shown that while protein markers of muscle damage are accurate, it is still not known when they appear, when they are cleared and how different exercise affects these markers. If these peptides are appearing due to protein breakdown, monitoring them over time may give a better indication as to when muscle damage is occurring.

Of the three biochemical energy pathways available to cells, amino acid degradation contributes the least under normal conditions. However, this pathway is of interest to endurance athletes as they aim to maximise fat loss but minimise muscle mass loss. With this in mind, key intermediate metabolic markers, alanine, glutamate, glutamine and arginine were investigated as indicators of this type of energy metabolism. Small changes in alanine and glutamate were observed in a similar pattern to glycolytic metabolism, but these were never above 2-fold compared to the baseline and arginine and glutamine remained at the same level across the data set. This data shows that amino acid energy metabolism is not happening at an appreciable level or above what would be expected in general, even during fasted recovery.

## 4. Materials and Methods

### 4.1. Subject Recruitment and Selection Criteria

Twelve participants (8 male, 4 female) were recruited on the basis of a high level of physical activity defined as completing a minimum 2.5 h intense physical activity per week. Physical activity was also assessed according to parameters outlined in the International Physical Activity Questionnaire (IPAQ), as well as the Physical Activity Readiness Questionnaire for Everyone (PAR-Q+). All participants included cycling as part of their regular training. Due to time constraints, only 7 participants completed the protocol (5 male, 2 female). Participant information is included in Table 1. All subjects gave their informed consent for inclusion before they participated in the study. The study was conducted in accordance with the Declaration of Helsinki, and the protocol was approved by the Ethics Committee of the University of Glasgow (200160106).

### 4.2. VO_2_ Max Determination Protocol

VO_2_ Max was determined using a CompuTrainer Lab Model 8002 Trainer (Racermate, Seattle, WA, USA) and a choice of road race bikes depending on fit. Prior to testing, participants were fasted for 2 h. Participants were initially required to cycle for 10 min at 50 W. After 10 min, a head mounted mouthpiece was fitted and remained in place for the duration of the protocol. After 1 min, at 50 W, participants were acclimatised to the mouthpiece, and power was increased every minute in 20–25 W increments (determined ad-hoc by tester) to the point of exhaustion. Expired air was collected during the final 2–3 min of testing via Douglas Bags and immediately analysed for O_2_ and CO_2_ concentration using a Servopro 1440 gas analyser (Servomex, East Sussex, United Kingdom).

### 4.3. Fasted Constant Load Session Phase

Prior to the session, participants were fasted for 12 h and were required to refrain from exercise for 24 h. On arrival, 5-min resting expired air samples was collected to determine baseline VO_2_ consumption, and RER was determined for baseline substrate utilisation. Once an RER of 0.70–0.85 was obtained, a cannula was inserted and participants were rested for 10 min. Upon completion of this resting period, a baseline blood sample was collected.

Immediately after baseline blood collection, participants began a 120-min cycling protocol using the same bike and CompuTrainer that was used to determine VO_2_ max. Participants cycled for 60 min at a power corresponding to 70% of the participants VO_2_ max. Upon completion, participants immediately began a 60-min flat course time trial at maximum exertion. The course was controlled by the CompuTrainer 3D software.

During the exercise protocol, blood samples were collected every 15 min (5 mL in a serum collection tube, 2 mL in an EDTA tube) along with 2-min breath samples in Douglas Air bags. Blood samples were centrifuged using a Hettich^®^ Universal 320R centrifuge (Sigma-Aldrich Company Ltd., Dorset, United Kingdom) for 10 min at 4000 rpm, and EDTA plasma was analysed for glucose and lactate concentrations using a YSI 2300 Stat Plus analyser (YSI Life Sciences, Yellow Springs, OH, USA). Serum and EDTA plasma aliquots were placed in 0.5 mL aliquot tubes and frozen at −80 °C for metabolomic analysis. Breath samples were analysed to determine RER. Participants were allowed to consume water at will and nothing else.

### 4.4. Fasted Rest Phase

Upon completion of the exercise protocol, participants were transferred to a seated position and remained at rest for 120 min. Blood samples were collected at 150, 180 and 240 min and breath samples were collected every 15 min, samples were processed as previously described for the constant load session. Participants were allowed to consume water at will and nothing else.

### 4.5. Metabolomic Analysis

All samples were analysed on a Thermo Scientific QExactive Orbitrap mass spectrometer running in positive/negative switching mode. This was connected to a Dionex UltiMate 3000 RSLC system (Thermo Fisher Scientific, Hemel Hempstead, United Kingdom) using a ZIC-pHILIC column (150 mm × 4.6 mm, 5 μm column, Merck Sequant, Gillingham, United Kingdom) The column was maintained at 30 °C and samples were eluted with a linear gradient (20 mM ammonium carbonate in water, A and acetonitrile, B) over 46 min at a flow rate of 0.3 mL/min as follows 0 min 20% A, 30 min 80% A, 31 min 92% A, 36 min 92% A, 37 min 20% A, 46 min 20% A.

The injection volume was 10 μL and samples were maintained at 5 °C prior to injection.

Mass spectrometry data was processed using a combination of XCMS 3.2.0 and MZMatch.R 1.0–4 [16,17]. Briefly, data was converted from Thermo proprietary raw files to the open format mzXML. Unique signals were extracted using the centwave algorithm and matched across biological replicates based on mass to charge ratio and retention time. These grouped peaks were then filtered based on relative standard deviation and combined into a single file. The combined sets were then filtered on signal to noise score, minimum intensity and minimum detections. The final peak set was then gap-filled and converted to text for use with IDEOM v18 [18]. The same processes were followed using PiMP [19]. The output used for IDEOM was also used for implementation of a generalised linear model (GLM) in the R coding language according to previous work by Rattigan et al. [20] using the lm function with *p*-values corrected using Benjamini-Hochberg.

## 5. Conclusions

This work has shown the utility of metabolomics for following changes in energy metabolism during fasted exercise and rest. Rather than following a small number of start and end point metabolites (such as glucose, lactate, 3-hydroxybutanoate, urea) the use of LC-MS has allowed for the investigation of a large number of intermediate metabolites in the three energy producing pathways, providing information on when each pathway is involved in energy production, even down to the precursors of fatty acid oxidation. This extends observations beyond simply determining if fat is being preferentially burned and shows a metabolomics approach can used to determine when this occurs and investigate if prolonged exercise is necessary, or if fasted rest can be substituted to maintain ketolytic energy production. Expansion of this could lead to development of novel training methods and exercise regimes to help endurance athletes and the general public alike, where weight management is of issue.

## Figures and Tables

**Figure 1 metabolites-10-00399-f001:**
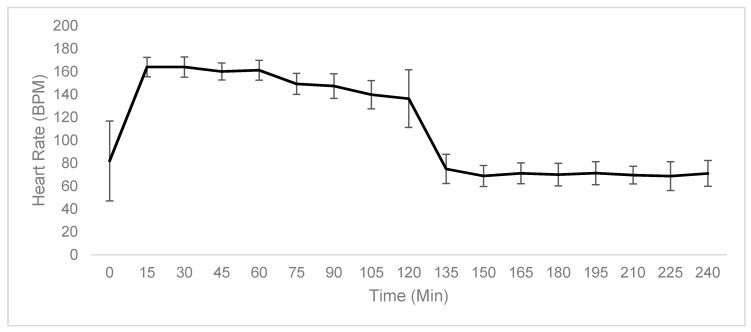
Average heart rate recorded at 15-min intervals during collection of breath samples. Small variations across subjects were observed during the exercise phases of the protocol. Upon rest, the average heart rate returned to the baseline and remained consistent across this phase.

**Figure 2 metabolites-10-00399-f002:**
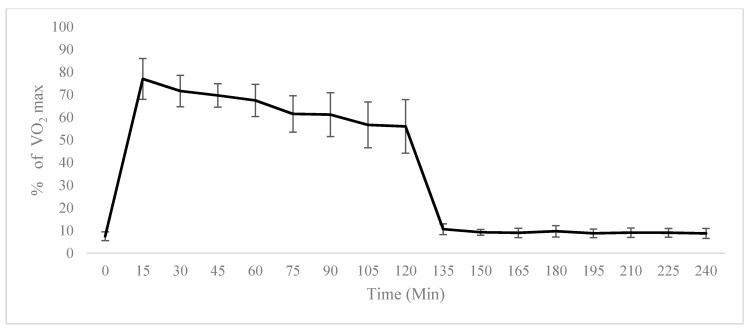
Average percentage of VO_2_ max based on calculation from respired air collection every 15 min. This data shows that effort was well conserved at approximately 70% of VO_2_ max for 60 min. Variation is seen across the time trail phase, with a general trend of slight reduction in effort. Variation increases towards the end of this section exercise phase, likely due to pacing differences in participants. Percentage of VO_2_ max is well conserved across the rest phase, indicating a similar level of exertion for all participants.

**Figure 3 metabolites-10-00399-f003:**
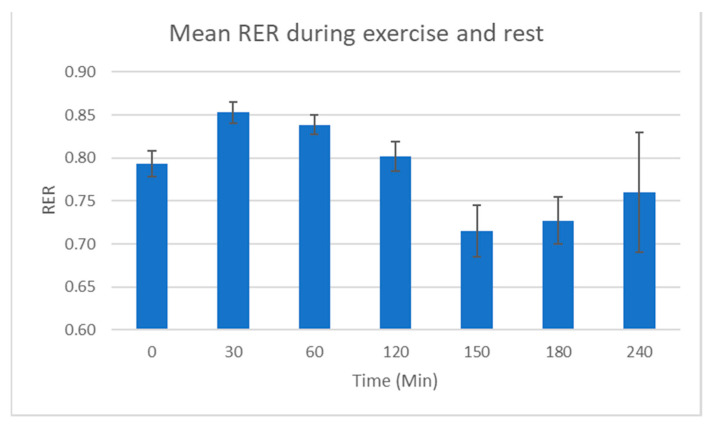
Average respiratory exchange ratio (RER) for all participants. Data suggests mainly glycolytic energy metabolism during the exercise phase of the protocol, then, an increase in fatty acid energy metabolism during rest with a trend back towards glycolytic metabolism at the end of the protocol.

**Figure 4 metabolites-10-00399-f004:**
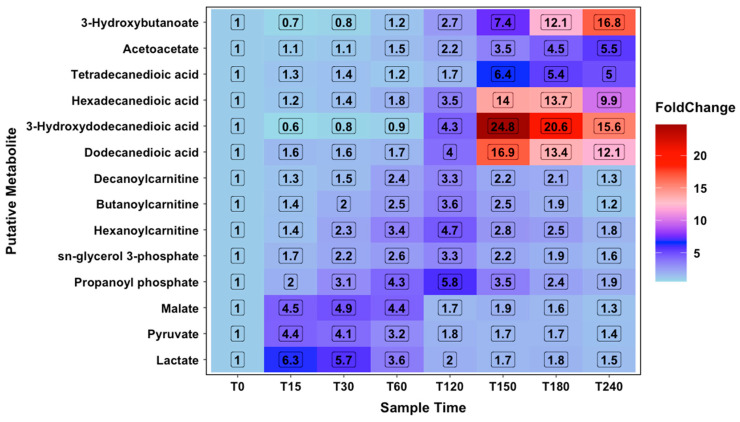
Metabolites showing fold changes across glycolytic and ketolytic energy metabolism pathways. Fold change values coloured from blue to red as fold change increases.

**Figure 5 metabolites-10-00399-f005:**
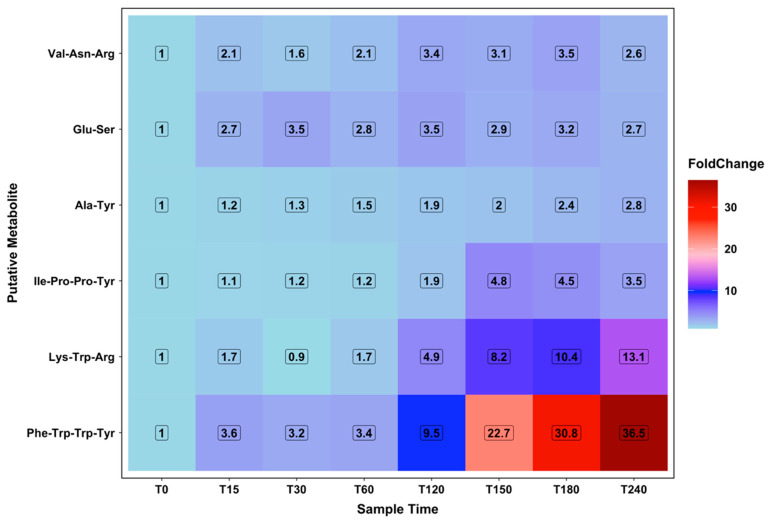
Metabolites putatively identified as peptides. Values coloured from blue to red as fold change increases.

**Table 1 metabolites-10-00399-t001:** Participant information including means and standard deviations for each value.

Subject ID	Gender	Age (Years)	Height (cm)	Weight (kg)	BMI (kg/m^2^)	Max. Oxygen Uptake (ml/kg/min)	Max. Power (Watts)
Metab-001	F	19	167.0	62.5	22.6	46.8	240
Metab-003	M	22	183.0	80.2	23.9	64.8	430
Metab-005	F	26	164.0	55.7	20.8	47.8	230
Metab-006	M	34	180.0	73.6	22.8	60.4	370
Metab-008	M	26	183.0	71.0	21.2	43.8	250
Metab-009	M	27	177.0	69.5	22.3	65.7	370
Metab-010	M	26	186.0	87.0	25.2	54.7	410
Mean		26	177.0	71.4	22.7	54.9	329
SD		5	8.5	9.7	1.5	9.0	86

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
