# Peer review of "Running on Empty: A Metabolomics Approach to Investigating Changing Energy Metabolism during Fasted Exercise and Rest"

_metabolites, 2020, doi:10.3390/metabo10100399_

Round 1

Reviewer 1 Report

The manuscript “Running on empty…” reports on an experiment in which metabolites were detected and quantified from blood samples from human participants undergoing an exercise time course. The authors noted several individual metabolites (and polypeptides) which change across the exercise time course and into recovery. The results suggest changes in glycolytic and ketogenic pathways during and immediately after exercise. While the sample size is small, the authors made use of multiple temporal measurements from the same subjects, which is powerful. Some positive aspects are listed below:

  1. Within subjects time course design is powerful relative to between subjects comparisons
  2. The LC-MS analysis seems very robust (from their supplementary data I see nice QC procedures using matrix blanks and pooled samples)
  3. Their supplementary data provides very detailed information on how the data was processed in their excel workbook. I think this is a nice step in reproducibility and transparency.
  4. The research question—more or less what metabolic changes can be noted in blood during and immediately after exercise—is very interesting and relevant to the readership of Metabolites

A few comments and suggestions:

  1. There are “error! Reference source not found” errors in lines 82 – 91 of section 2.1.
  2. Figure 1 shows the average (over subjects) for heart rate and Figure 2 shows the same for V02 max. Would it be possible to show these as a time-course within each individual? I think this would be more informative showing the individual trajectories (and the consistency or lack there of between individuals).
  3. There are two figures labeled as figure 1. This may have something to do with the “error! Reference source not found” errors.
  4. Providing more details on the GLM would be beneficial. It is hard to understand the form of the model as is. It seems like the GLM included an intercept and time variable only to allow for calculating average fold change across time. It is not clear here how the correlation in the measurements is dealt with. There is likely correlation in the measurements from each subject. The authors do mention not including an interaction term due to missing time points. However, it seems like the best way to deal with this would be to use a mixed effects model rather than a fixed effects model. Then the dependence between measurements could be accounted for, leading to more reliable (and accurate) inferences. If the authors are using R, the lme4 package could be used to do this (with a easy learning curve for implementing such a model).
  5. The authors report FDR = 0.00 in many instances. It is nice that the authors are reporting measures of statistical significance that preserve the false discovery rate rather than p-values. However, I think using (for example) “q = 0.0001” or “adjusted p = 0.0001” rather than “FDR = 0.0001” makes more sense. Q-values or Benjamini Hochberg adjusted p-values use empirical procedures to attempt to preserve the FDR at (or below) a specific value if the specific q-value is set at that threshold—so it is strange to state something like FDR = XX as the FDR is unknown. Also, I would suggest leaving at least one significant digit or using scientific notation for “FDR = 0.00” since it is not zero. For example q = 0.001 or q = 0.0001 or q = 0.00001 or q < 0.00001. This would also give some sense of what percentage of similar results called “significant” would be false discoveries if the q-value was the threshold. Finally, it would be nice to clarify what type of FDR preserving procedure is used here as there are many kinds (Benjamini Hochberg, Storey q-value, local FDR control procdures).

Author Response

We would like to thank the reviewer for their comments. Please find attached our responses to the reviewers points/questions:

  1. here are “error! Reference source not found” errors in lines 82 – 91 of section 2.1.

We thank the reviewer for pointing out this error, it has been corrected.

  1. Figure 1 shows the average (over subjects) for heart rate and Figure 2 shows the same for V02 max. Would it be possible to show these as a time-course within each individual? I think this would be more informative showing the individual trajectories (and the consistency or lack there of between individuals).

The general trend is the same across subjects. There are other factors that influence heart rate, such as temperature, hydration etc. and these data are not available. As these HR data were recorded at the time points stated, rather than continuously, we feel that splitting the data in this case would not provide any additional information over and above what the current figures provide. However, we do agree that individualised time courses would be beneficial for future work and thank the reviewer for pointing this out. As such, we will endeavour to record this information in future experiments and provide it with future publication.

  1. There are two figures labeled as figure 1. This may have something to do with the “error! Reference source not found” errors.

We thank the reviewer for pointing out this error, it has been corrected.

  1. Providing more details on the GLM would be beneficial. It is hard to understand the form of the model as is. It seems like the GLM included an intercept and time variable only to allow for calculating average fold change across time. It is not clear here how the correlation in the measurements is dealt with. There is likely correlation in the measurements from each subject. The authors do mention not including an interaction term due to missing time points. However, it seems like the best way to deal with this would be to use a mixed effects model rather than a fixed effects model. Then the dependence between measurements could be accounted for, leading to more reliable (and accurate) inferences. If the authors are using R, the lme4 package could be used to do this (with a easy learning curve for implementing such a model).

We thank the reviewer for this comment. The GLM is described in the referenced paper and as such we did not include a large amount of information. Some more information has been provided that will hopefully satisfy the reviewers point regarding the model and we have included the method of correction (Benjamini-Hochberg). The model performs an ANOVA for each detected signal across the relevant factors and tests if the result is linear (no change across the factor). This gives a p-value that is then corrected using a Benjamini-Hochberg correction. Those signals returning a significant corrected p-value were plotted to investigate the shape of the difference across the detected signal and interpreted based on the biology present. We have expanded this description in the text.

We agree that there may be interesting correlations across the subjects, however due to the wide variation in signal intensity, lack of repeat measurements and small number of subjects, we feel that this work would likely not provide a robust enough data set to perform such an analysis.

  1. The authors report FDR = 0.00 in many instances. It is nice that the authors are reporting measures of statistical significance that preserve the false discovery rate rather than p-values. However, I think using (for example) “q = 0.0001” or “adjusted p = 0.0001” rather than “FDR = 0.0001” makes more sense. Q-values or Benjamini Hochberg adjusted p-values use empirical procedures to attempt to preserve the FDR at (or below) a specific value if the specific q-value is set at that threshold—so it is strange to state something like FDR = XX as the FDR is unknown. Also, I would suggest leaving at least one significant digit or using scientific notation for “FDR = 0.00” since it is not zero. For example q = 0.001 or q = 0.0001 or q = 0.00001 or q < 0.00001. This would also give some sense of what percentage of similar results called “significant” would be false discoveries if the q-value was the threshold. Finally, it would be nice to clarify what type of FDR preserving procedure is used here as there are many kinds (Benjamini Hochberg, Storey q-value, local FDR control procdures).

This is a very good point. We have replaced FDR with benjamini corrected p-values and used scientific notation.

Again, we would like to thank the reviewer for their comments and questions/corrections, it is greatly appreciated.

Reviewer 2 Report

This manuscript aims to prove the concept of using untargeted metabolomics screening to analyse use of energy-yielding substrates during exercise to exhaustion. A range of glycolytic and lipolytic metabolites are measured in blood along with calculations of RER to establish predominant substrate utilisation. The concept is an interesting and topical one and will be of interest to researchers in metabolism well as to those concerned with health management.

Major Comments.

Some adjustment of view was required for me on reading this manuscript. Many of the conclusions drawn depend on measurements of metabolites in the bloodstream which would normally only be found in the intracellular compartment. Thus, we might expect any increase in concentrations of malate, pyruvate, etc to indicate tissue damage and leakage into the plasma. In reality, there is little indication that muscle damage has occurred to any great degree since we apparently do not see many branched chain amino acids or peptides containing them listed in the peptide heat map. However, perhaps the authors could acknowledge that measurements of highly unstable metabolites in the plasma may produce ambiguous interpretations.

Moreover, perhaps more could be done by the authors to indicate that the roles of the peptides found are unclear and that evidence of tissue damage is difficult to distinguish from putative signalling roles. Have the authors conducted a through search to ascertain whether any signalling roles have previously been suggested for these or similar peptides?

Section 2.6 : I am unclear why measurements have been made of glutamate levels but not of those of glutamine. Glutamine would normally be the vehicle of nitrogen transport in the blood to indicate increased amino acid metabolism in muscle tissue (which I think must occur under these conditions), rather than glutamate.

On the same theme, the variations in RER among the subjects probably indicate differing levels of glycogen in their muscles which may result from different levels of training. This ought to be acknowledged in the text. It should be noted that, on moving to the ketolytic phase, a carbohydrate source would be required to ensure that increased ketone body metabolism proceeded efficiently. In the fasted state, this is likely to derive from muscle proteins once glycogen deposits have been depleted.

Minor comments

There are some indications that this manuscript has been submitted before proof-reading is entirely complete.

Lines 83-91 : References to Figures need to be added.

Line 262 : again, the appropriate reference is missing.

Line 223 : contains an unfinished statement.

Line 232 : I suspect the authors don’t really mean to refer to a “protein kinase”.

Author Response

We would like to thank the reviewer for their comments. Please find attached our responses to the reviewers points/questions:

Major comments

  1. Some adjustment of view was required for me on reading this manuscript. Many of the conclusions drawn depend on measurements of metabolites in the bloodstream which would normally only be found in the intracellular compartment. Thus, we might expect any increase in concentrations of malate, pyruvate, etc to indicate tissue damage and leakage into the plasma. In reality, there is little indication that muscle damage has occurred to any great degree since we apparently do not see many branched chain amino acids or peptides containing them listed in the peptide heat map. However, perhaps the authors could acknowledge that measurements of highly unstable metabolites in the plasma may produce ambiguous interpretations.

We thank the reviewer for this discussion and we agree that many of these metabolites would be expected to be found within the intracellular compartment. There is a body of work detailing their analysis in plasma and serum. As muscle breakdown was investigated as a source of energy metabolism we are confident, based on this data, that this was minimal/in line with what may be expected from fasted exercise in general. We have included an acknowledgement of this within the text as suggested.

  1. Moreover, perhaps more could be done by the authors to indicate that the roles of the peptides found are unclear and that evidence of tissue damage is difficult to distinguish from putative signalling roles. Have the authors conducted a through search to ascertain whether any signalling roles have previously been suggested for these or similar peptides?

This is a great point and we have modified the text to more firmly indicate that their role is ambiguous. During the analysis of the data the suggested sequences of these peptides were checked against known peptide hormones associated with diet and hunger, such as Ghrelin, to determine if they may be fragments of these, however due to their short length this was not successful. It should also be noted that this particular analysis is more geared towards small, soluble molecules. This means that additional information from larger peptides that may have helped determine their origin is lost. This is certainly something that we would consider for a larger experiment when sample numbers are increased and a greater range of analytical platforms are used.

  1. Section 2.6 : I am unclear why measurements have been made of glutamate levels but not of those of glutamine. Glutamine would normally be the vehicle of nitrogen transport in the blood to indicate increased amino acid metabolism in muscle tissue (which I think must occur under these conditions), rather than glutamate.

We thank the reviewer for bringing this to our attention. Glutamine was measured in the analysis and follows a similar trend to glutamate. Alanine, glutamate and arginine were chosen as they are well observed on this system and were used to investigate if there were changes observed in the urea cycle. We have included Glutamine in the text and noted its role.

  1. On the same theme, the variations in RER among the subjects probably indicate differing levels of glycogen in their muscles which may result from different levels of training. This ought to be acknowledged in the text. It should be noted that, on moving to the ketolytic phase, a carbohydrate source would be required to ensure that increased ketone body metabolism proceeded efficiently. In the fasted state, this is likely to derive from muscle proteins once glycogen deposits have been depleted.

With regard to RE we agree with the reviewer on this observation and thank them for noticing this omission. We have updated the text to reflect this.

In regard to carbohydrate source in the fasted state, the data does not suggest a large increase in protein catabolism, however this method is not quantitative and as such it is difficult to determine the contribution of each different component of energy metabolism at the different time points, just if there has been in increase in molecules associated with it. We also do not have any relevant data to assess how large a shift in protein metabolism related signals would be required in this data for us to determine the relative contribution of the different energy sources. A comparative data set of non-fasted individuals would aid with this, but this is beyond the scope of this particular dataset. As such, we have not drawn any conclusions in this area. The conclusions have been expanded to cover this point.

Minor comments

There are some indications that this manuscript has been submitted before proof-reading is entirely complete.

Lines 83-91 : References to Figures need to be added.

Line 262 : again, the appropriate reference is missing.

Line 223 : contains an unfinished statement.

Line 232 : I suspect the authors don’t really mean to refer to a “protein kinase”.

We thank the reviewer for bringing these to our attention and have corrected/amended as appropriate.

Again, we would like to thank the reviewer for taking the time to assess the work and the valuable points they have provided, it is greatly appreciated.

Round 2

Reviewer 1 Report

The authors addressed the comments / suggestions that I had made in the first review.

I believe the work is very interesting (the question of how the plasma metabolome changes during and after exercise), and the analysis is described very well and seems robust. I believe it would be a good contribution to the journal and appreciated by the readers.

Author Response

We would like to thank the reviewer for their words and for taking the time to review manuscript.

Reviewer 2 Report

As a proof of concept, this study makes a contribution to the field of exercise science.  The authors establish the feasibility of measuring blood metabolites at intervals during rest and exercise and observing metabolic changes.  As such, many successful observations are made.  As regards the identification of metabolites, there is very little, if anything , in this manuscript that has not been very well-established previously.  The authors occasionally stray into the territory of making claims for their data which cannot be substantiated.  I feel this manuscript ought to be re-orientated to emphasise the procedural aspects rather than trying to make a contribution to knowledge of metabolic control.

Specific comments

Figure 1 legend : The legend needs to be re-written.  The fact that values ”remain consistent across this phase” does not in any way indicate a “similar level of training and exertion” throughout the group.  The magnitude of the standard deviation, which indicates the variation of individual data points around the mean, would indicate this.

Figure 3 and its analysis : The interpretation of the data presented in this figure really does overstate the case.  Whereas the decrease in the RER during the subsequent rest phase compared with the preceding exercise phase does indeed indicate that metabolism may well be more carbohydrate-centred during exercise than at rest, it is not possible to say that it is “mainly glycolytic” during exercise.  Even during the exercise phase, values of the RER are between 0.8 and 0.85, indicating that there may be a substantial contribution of fat oxidation to energy generation.

L203-205 : this amendment needs further attention.  Pyruvate is indeed an intracellular metabolite and is notoriously unstable in the plasma.  Lactate, by contrast, is much more stable in the blood and is a vehicle for carbon transport between muscle and liver.  Levels of lactate would thus be expected to increase in the  blood during intensive exercise as a result of the operation of the Cori cycle.

Author Response

"As a proof of concept, this study makes a contribution to the field of exercise science.  The authors establish the feasibility of measuring blood metabolites at intervals during rest and exercise and observing metabolic changes.  As such, many successful observations are made.  As regards the identification of metabolites, there is very little, if anything , in this manuscript that has not been very well-established previously.  The authors occasionally stray into the territory of making claims for their data which cannot be substantiated.  I feel this manuscript ought to be re-orientated to emphasise the procedural aspects rather than trying to make a contribution to knowledge of metabolic control."

We thank the reviewer for this discussion and hope the changes made to the manuscript will address any concerns. As the reviewer has noted, the metabolism discussed has been previously established and this work demonstrates the application of LC-MS metabolomics methods to measuring these metabolites. The data presented shows that the metabolites discussed respond as would be expected and allows the following of key intermediate metabolites, such as acyl-carnitines. We believe that the discussion of, for example the small peptide signals seen within the data, also highlights the benefit of this method, while not drawing specific conclusions on their role and origin which is not possible from this data.

"Specific comments

Figure 1 legend : The legend needs to be re-written.  The fact that values ”remain consistent across this phase” does not in any way indicate a “similar level of training and exertion” throughout the group.  The magnitude of the standard deviation, which indicates the variation of individual data points around the mean, would indicate this."

We thank the reviewer for pointing this out and agree that this overstates the data in the figure. This is discussed elsewhere and as such has been removed from the legend.

"Figure 3 and its analysis : The interpretation of the data presented in this figure really does overstate the case.  Whereas the decrease in the RER during the subsequent rest phase compared with the preceding exercise phase does indeed indicate that metabolism may well be more carbohydrate-centred during exercise than at rest, it is not possible to say that it is “mainly glycolytic” during exercise.  Even during the exercise phase, values of the RER are between 0.8 and 0.85, indicating that there may be a substantial contribution of fat oxidation to energy generation."

We thank the reviewer for pointing this out and agree this overstates the data here. These observations are made elsewhere and tie in with the data observed in other figures, particularly figure 4, where they are more appropriate. The legend has been re-written to better reflect the data presented in the figure.

"L203-205 : this amendment needs further attention.  Pyruvate is indeed an intracellular metabolite and is notoriously unstable in the plasma.  Lactate, by contrast, is much more stable in the blood and is a vehicle for carbon transport between muscle and liver.  Levels of lactate would thus be expected to increase in the  blood during intensive exercise as a result of the operation of the Cori cycle."

We thank the reviewer for this comment and have readdressed this section. Lactate follows the expected trend for intensive exercise, as can be seen in figure 4, as does pyruvate and malate. The instability of metabolites could affect absolute concentrations and if quantification methods were being used we would have taken this into account in the experimental design. In this case we are looking at relative levels and, based on the metabolites reported in the manuscript following long established patterns, with glycolytic metabolites being raised under intensive exercise and ketolytic metabolites increasing under the time trial and recovery phases, we believe this fairly represents the data presented.

Round 3

Reviewer 2 Report

More could still be done to emphasise the methodological contribution made by this study rather than the metabolic findings.